# K-Sil: Silhouette-Driven Instance-Weighted Clustering

## Abstract

Clustering is a fundamental unsupervised learning task with numerous applications across diverse fields. Popular algorithms such as $k$-means often struggle with outliers or imbalances, leading to distorted centroids and suboptimal partitions. We introduce K-Sil, a silhouette-guided refinement of the $k$-means algorithm that weights points based on their silhouette scores, prioritizing well-clustered instances while suppressing borderline or noisy regions. The algorithm emphasizes user-specified silhouette aggregation metrics: macro-, micro-averaged or a combination, through self-tuning weighting schemes, supported by appropriate sampling strategies and scalable approximations. These components ensure computational efficiency and adaptability to diverse dataset geometries. Theoretical guarantees establish centroid convergence, and empirical validation on synthetic and real-world datasets demonstrates statistically significant improvements in silhouette scores over $k$-means and two other instance-weighted $k$-means variants. These results establish K-Sil as a principled alternative for applications demanding high-quality, well-separated clusters.

## 1 Introduction

Clustering, the process of organizing data into meaningful groups, is a cornerstone of unsupervised learning with broad applications in pattern recognition and data analysis Jain et al. (1999). Among clustering algorithms, $k$-means MacQueen (1967) remains widely adopted due to its simplicity and scalability. However, its susceptibility to outlier distortion and poor separation in imbalanced clusters frequently results in centroids skewed by noise, producing partitions that misrepresent the underlying data structure, especially in cases with small or overlapping groups Pavlopoulos et al. (2025). While weighted variants mitigate some limitations, they often lack a principled mechanism to prioritize high-confidence points or suppress unreliable ones Xu & Wunsch (2005), limiting their ability to better capture intrinsic data geometries.

To address these challenges, we introduce K-Sil, a clustering refinement that integrates silhouette scores Rousseeuw (1987); Dudek (2020), a per-point measure balancing intra-cluster cohesion and nearest-cluster separation, directly into the centroid update process. Unlike standard $k$-means, which treats all points uniformly, K-Sil dynamically weights each point based on its silhouette within its assigned cluster: well-clustered points exert greater influence on centroid updates, while borderline or noisy points are systematically downweighted. This local, interpretable reweighting steers centroids toward stable cores while retaining the familiar Lloyd-style loop and practical runtime via a lightweight silhouette approximation and objective-aware sampling.

Overall, the key contributions of this work are: *(I)* we define clustering objectives guided by silhouette-based aggregation metrics, including the macro-averaged silhouette for assessing cluster-level quality, the micro-averaged silhouette for evaluating average point-wise cohesion, and convex combinations to balance both, enabling users to prioritize distinct aspects of partition quality; *(II)* we develop auto-tuned per-cluster weighting schemes that emphasize high-confidence regions and suppress unreliable assignments, leveraging either silhouette magnitudes (absolute scores) or ranks (relative ordering); *(III)* we reduce the computa-

tional costs of silhouette calculation using objective-aware sampling—per-cluster sampling for macro objectives and uniform sampling for micro—together with a centroid/dispersion-based approximation that avoids $O(n^2)$ pairwise distances and keeps each iteration near $O(nk)$; *(IV)* we establish finite convergence under standard cluster regularity assumptions and validate K-Sil empirically, showing statistically significant improvements in macro/micro silhouette over $k$-means and other instance-weighted $k$-means variants across synthetic and real-world datasets. Our code is publicly available at: https://anonymous.4open.science/r/ksil-ICLR.

## 2 RELATED WORK

Clustering methods are often categorized by their strategy: partitioning (e.g., $k$-means MacQueen (1967)), density-based (e.g., DBSCAN Ester et al. (1996)), hierarchical Müllner (2011), model-based (e.g., GMMs Reynolds (2009)), and graph/spectral approaches Ng et al. (2002). These vary in assumptions, metrics, and scalability. Our work focuses on improving partition-based clustering, specifically $k$-means, by incorporating instance-level weighting guided by internal validation signals. $k$-means MacQueen (1967) partitions data into $k$ disjoint clusters by minimizing intra-cluster variance through iterative centroid updates. Though efficient, it is sensitive to initialization and local optima. Methods such as $k$-means++ Arthur & Vassilvitskii (2007) and AFKM Bachem et al. (2016) improve stability, while global optimization strategies like genetic $k$-means Krishna & Murty (1999) and global $k$-means Likas et al. (2003); Vardakas & Likas (2022) attempt to escape local minima. U-$k$-means Sinaga & Yang (2020) further extends the approach by estimating $k$ during clustering. However, these variants generally assume uniform instance influence (and often uniform feature importance), leaving $k$-means vulnerable to outliers, noise, and imbalanced data.

Weighted extensions address these limitations by adjusting the influence of features or instances. For example, WK-Means Huang et al. (2005) and EWKM Jing et al. (2007) assign feature weights based on compactness or entropy, while AWA Chan et al. (2004) uses variance-based weighting. Instance-based methods, such as weight-balanced $k$-means Borgwardt et al. (2013) and LOF-based approaches (LOFKMeans) Moggridge et al. (2020), incorporate external or density-aware signals, but often lack adaptive, point-level iterative refinement.

Internal metrics like the silhouette coefficient Rousseeuw (1987); Arbelaitz et al. (2013), traditionally used for evaluation, have increasingly been employed to guide clustering. Some methods reassign points iteratively to improve silhouette Bombina et al. (2024), while others such as WKBSC Lai et al. (2024) integrate silhouette into the objective function. Most of these approaches focus on the micro-averaged silhouette, overlooking macro-averaged alternatives Pavlopoulos et al. (2025) that better capture structure in imbalanced data. Additionally, fast approximations of silhouette may misestimate cohesion or separation, and global optimization based solely on silhouette Lai et al. (2024); Batool & Hennig (2021) can lead to overfitting, amplifying noise or local irregularities. These issues underscore the need for more localized, interpretable uses of silhouette, especially at the instance level, to improve clustering structure without distorting it.

## 3 METHODOLOGY

Let $X = \{x_1, x_2, \ldots, x_n\}$ be a dataset consisting of $n$ data points in the metric space $(\mathbb{R}^d, \|\cdot\|)$, where $\|\cdot\|$ denotes the $\ell_2$-norm. Given a partition of $X$ into clusters $\{C_1, \ldots, C_k\}$, the silhouette score for a point $x_i \in C_j$, $s(x_i)$ quantifies the quality of its cluster assignment by comparing its average intra-cluster distance $a(x_i)$ to its minimum average inter-cluster distance $b(x_i)$:

$$s(x_i) = \frac{b(x_i) - a(x_i)}{\max\{a(x_i), b(x_i)\}} \in [-1, 1], \tag{1}$$

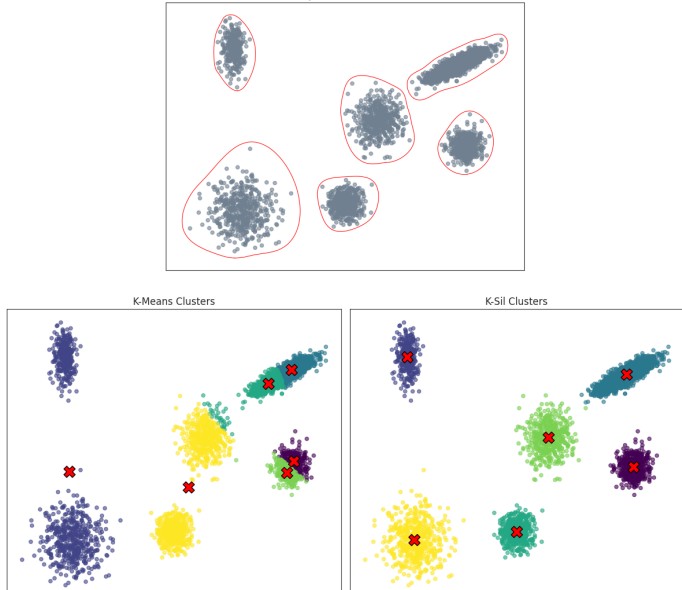

Figure 1: Assignments (and centroids marked in red) produced by K-Means (bottom left) and K-Sil (bottom right), both initialized with the same centroids, applied to a synthetic dataset (top). Under the same initialization, K-Sil's weighting steers centroids toward well-separated cores and away from overlaps, yielding a cleaner partition than K-Means in this run.

$$a(x_i) = \frac{1}{|C_j| - 1} \sum_{\substack{x_h \in C_j \\ i \neq h}} \|x_i - x_h\|, \quad b(x_i) = \min_{h \neq j} \left\{ \frac{1}{|C_h|} \sum_{x_m \in C_h} \|x_i - x_m\| \right\}.$$

To evaluate overall clustering quality beyond individual silhouette values and obtain a single summary-metric, we aggregate silhouette scores at the dataset level Pavlopoulos et al. (2025). We define the **micro-averaged silhouette score** $S_{\mathrm{m}}$ (the mean of the silhouette scores over all data points) that prioritizes the average quality of individual assignments, particularly effective for datasets with balanced cluster sizes, and the **macro-averaged silhouette score** $S_{\mathrm{M}}$ (the average of the per-cluster mean silhouette scores) emphasizing performance across clusters, independent of size, suitable for evaluating partitions with imbalanced clusters or when uniform cluster quality is critical:

$$S_{\mathrm{m}} = \frac{1}{n} \sum_{x_i \in X} s(x_i), \quad S_{\mathrm{M}} = \frac{1}{k} \sum_{j=1}^{k} \frac{1}{|C_{\mathrm{j}}|} \sum_{x_i \in C_{\mathrm{j}}} s(x_i). \tag{2}$$

**Clustering Objective** The K-Sil algorithm partitions $X$ into $k$ disjoint clusters $\{C_1, C_2, \ldots, C_k\}$ ($k < n$), with corresponding centroids $\{\mu_1, \mu_2, \ldots, \mu_k\} \subset \mathbb{R}^d$, through an iterative refinement procedure that maximizes the selected silhouette aggregation objective $S$, either $S_{\mathrm{M}}$, $S_{\mathrm{m}}$ or a convex combination $\alpha S_{\mathrm{m}} + (1-\alpha)S_{\mathrm{M}}$. At each iteration $t$, clusters $C_j$ are denoted by

$$C_j^{(\mathrm{t})} = \{x_i \in X : \|x_i - \mu_j^{(\mathrm{t})}\| \leq \|x_i - \mu_h^{(\mathrm{t})}\| \; \forall h \in \{1, \ldots, k\}\},$$

where $\mu_j^{(\mathrm{t})}$ denotes the centroid of cluster $C_j^{(\mathrm{t})}$. Also, let $S^{(t)}$ and $s_i^{(t)}$ denote the value of the objective and the silhouette score of data point $x_i$ at iteration $t$.

**Initialization** The algorithm obtains the initial clusters by selecting centroids through a single $k$-means iteration. The centroids are selected via either random initialization or the k-means++ method, which identifies well-separated starting points.

### 3.1 ITERATIVE REFINEMENT AND CENTROID UPDATES

Once the initial clusters are established, the algorithm proceeds with the iterative refinement. At each iteration $t$, the following steps are executed (detailed outline in Appendix A.1).

**Silhouette Scores Computation**  The algorithm iteratively refines clusters via silhouette scores computed for all data points within each cluster. For large datasets, where computing exact silhouette scores becomes computationally prohibitive due to $O(n^2)$ pairwise distance calculations, the algorithm offers two options for scalability. First, objective-aligned sampling reduces the number of points evaluated: Uniform sampling preserves proportional cluster representation for the micro-averaged objective $S_{\mathrm{m}}$, while per-cluster sampling enforces equal representation for the macro-averaged objective $S_{\mathrm{M}}$. Second, a silhouette approximation that avoids pairwise distance computations by estimating intra-cluster and inter-cluster distances using cluster sizes, centroids and within clusters sum of squares ($SS$), effectively reducing computational complexity to $O(nk)$ or, when sampling is enabled and the number of sampled points is $m$ ($< n$), to $O(mk)$. Specifically, for each point $x_i \in C_j$, with $j \in \{1, \ldots, k\}$, the approximate $\tilde{a}(x_i)$ and $\tilde{b}(x_i)$ for the silhouette computation are defined as:

$$\tilde{a}(x_i) = \sqrt{\frac{|C_{\mathrm{j}}| \cdot \|x_{\mathrm{i}} - \mu_{\mathrm{j}}\|^2 + SS_{C_{\mathrm{j}}}}{|C_{\mathrm{j}}| - 1}}, \quad \tilde{b}(x_{\mathrm{i}}) = \min_{h \neq j} \sqrt{\|x_{\mathrm{i}} - \mu_{\mathrm{h}}\|^2 + \frac{SS_{C_{\mathrm{h}}}}{|C_{\mathrm{h}}|}},$$

$$\tilde{s}(x_i) = \frac{\tilde{b}(x_i) - \tilde{a}(x_i)}{\max\left\{\tilde{a}(x_i), \tilde{b}(x_i)\right\}}, \tag{3}$$

where $SS_{C_j} = \sum_{x \in C_j} \|x - \mu_j\|^2$. These approximations improve upon the commonly used centroid-distance proxy that can be overly simplistic, by incorporating cluster variability. The approximated $\tilde{a}(x_i)$ captures both point-to-centroid distance and cluster spread, yielding a more accurate intra-cluster estimate. Likewise, $\tilde{b}(x_i)$ reflects distance to other centroids adjusted for their dispersion, mitigating overestimation in high-variance clusters. While absolute scores may differ slightly from exact values, their relative ordering, and overall silhouette behavior, remains consistent (see also Appendix Tables 3 and 4 for correlation analysis with exact scores and comparisons with simpler proxies).

**Instance Weighting**  Within each cluster $C_j^{(\mathrm{t})}$, (sampled) points are assigned weights based on their silhouette scores to prioritize well-clustered regions and de-emphasize low-silhouette ones during centroid updates. Two weighting schemes are employed, each with distinct advantages depending on cluster homogeneity:

**Power Weighting Scheme**: For $x_i \in C_j^{(\mathrm{t})}$, the weight $w_i^{(\mathrm{t})} = w(x_i)$ is defined as:

$$w_i^{(\mathrm{t})} = \left[\frac{s^{(\mathrm{t})}(x_i) - s_{\min}(C_j^{(\mathrm{t})}) + \epsilon}{\mathrm{Median}\left(\left\{(s^{(\mathrm{t})}(x_h) - s_{\min}(C_j^{(\mathrm{t})}) + \epsilon) \mid x_h \in C_j^{(\mathrm{t})}\right\}\right)}\right]^p, \tag{4}$$

where $s_{\min}(C_j^{(\mathrm{t})})$ is the minimum silhouette score in $C_j^{(\mathrm{t})}$, $\epsilon$ is a small constant ensuring numerical stability and $p > 0$ is a weight sensitivity parameter controlling the weight contrast. This scheme amplifies deviations from the minimum silhouette, median-scaled, making it effective for homogeneous clusters where absolute silhouette differences reliably distinguish core points from noise.

**Exponential Weighting Scheme**: For $x_i \in C_j^{(\mathrm{t})}$, the weight $w_i^{(\mathrm{t})} = w(x_i)$ is defined as:

$$w_i^{(\mathrm{t})} = \exp\left[-p \cdot \frac{\mathrm{rank}(s^{(\mathrm{t})}(x_i)) - \mathrm{Median}\left(\left\{\mathrm{rank}(s^{(\mathrm{t})}(x_h)) \mid x_h \in C_j^{(\mathrm{t})}\right\}\right)}{\mathrm{rank}(s_{\min}(C_j^{(\mathrm{t})})) / 2}\right], \tag{5}$$

where $\text{rank}(s^{(t)}(x_i))$ is the descending (dense) rank (points with higher silhouette scores receives lower ranks) of the $s^{(t)}(x_i)$ among the other silhouette scores of points in $C_j^{(t)}$, where tied scores share the same rank, $s_{\min}(C_j^{(t)})$ is the maximum dense rank, corresponding to the rank of the cluster's minimum silhouette score and $p > 0$ is a weight sensitivity parameter that controls the exponential decay rate. This scheme is robust for heterogeneous clusters (e.g., variable densities) and compatible with the approximation strategy, as it prioritizes relative ranks over absolute scores.

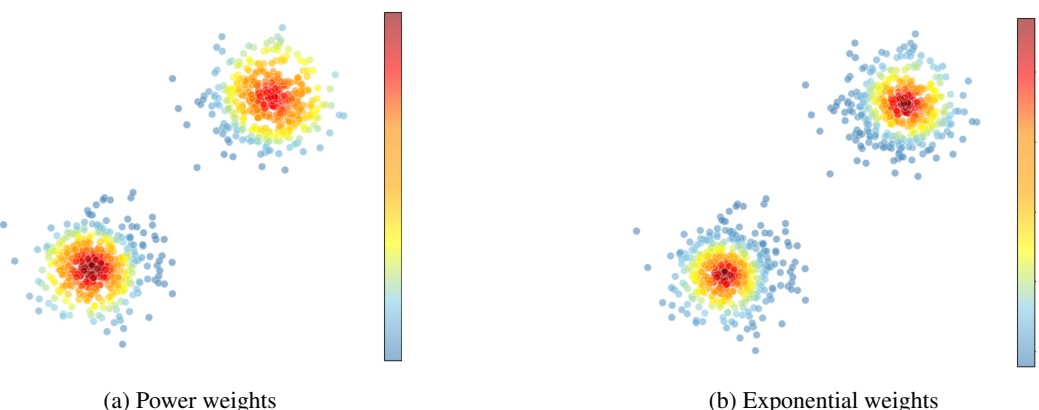

(a) Power weights            (b) Exponential weights

Figure 2: Normalized instance weights from the power (left) and exponential (right) weighting schemes (each configured with a weight sensitivity factor) on synthetic data of two Gaussian clusters.

**Centroid Updates and Convergence** After assigning weights to points in each cluster, the algorithm updates centroids to reflect the weighted influence of points. This weighted mean ensures centroids shift toward regions of high silhouette scores, enhancing cluster compactness and separation (see also Appendix A.3 for the empty-cluster re-initialization strategy). Finally, the algorithm terminates when the average centroid movement falls below a threshold $\tau$ $\left( \frac{1}{k} \sum_{j=1}^{k} \| \mu_j^{(t+1)} - \mu_j^{(t)} \| < \tau \right)$ or a maximum iteration limit is reached, and the partition (centroids, labels) with the highest observed silhouette objective score $S^*$ across all iterations is retained [1] (see Appendix A.4 for the highest silhouette retained partition).

### 3.2 WEIGHT-SENSITIVITY

K-Sil's weighting schemes amplify the influence of high-silhouette points while suppressing low-silhouette ones, controlled by a weight-sensitivity parameter $p$. This parameter modulates the contrast between weights: points with silhouette scores above the cluster median receive weights $> 1$, increasing their impact on centroid updates, while those below receive weights $< 1$, reducing their influence. Higher $p$ values sharpen this contrast, emphasizing well-clustered points; lower values soften it, retaining borderline regions. $p$ can be auto-tuned via grid search to maximize the chosen silhouette objective ($S_M$, $S_m$, or a combination), or set manually for efficiency in time-sensitive scenarios (silhouette score variation across sensitivity values is depicted in the following plots).

---

[1]The centroid movement threshold is typically sufficient for convergence. The maximum iteration limit serves only as a practical safeguard and is rarely reached in practice.

(a) MICE PROTEIN

(b) GLASS

(c) 20 NEWSGROUPS Subset

Figure 3: K-Sil's performance on $S_M$ across the weight-sensitivity parameter $p$ (0–20). At $p=0$, all instance weights equal 1 and K-Sil coincides with standard $k$-means; the value shown at $p=0$ is therefore the $k$-means partition with the highest silhouette encountered during its iterations. Blue: power weighting; red: exponential weighting. Evaluated on the Mice Protein, Glass, and 20 Newsgroups (subset) datasets.

## 4    THEORETICAL ANALYSIS

We establish theoretical guarantees for the K-Sil algorithm, emphasizing centroid convergence in finite time under assumptions (see Appendix B for detailed derivations).

### 4.1    CLUSTER PARTITIONING BY SILHOUETTE SCORES

Given that $C_j^{(t)}$ represents the $j$-th cluster at iteration $t$ with centroid $\mu_j^{(t)}$, we analyze its structure by partitioning points based on their silhouette scores.

Using $m_j^{(t)} = \text{Median}\left(\left\{s^{(t)}(x_i) \mid x_i \in C_j^{(t)}\right\}\right)$, the median silhouette score of all points in $C_j^{(t)}$, we partition each cluster into two key subsets to analyze the algorithm's behavior: $H_j^{(t)}$, the subset of high-silhouette (core) points that are strongly aligned with their cluster, and drive centroid updates due to their high weights ($> 1$), and $L_j^{(t)}$, the subset of low-silhouette (peripheral or borderline) points that lie farther from the cluster's core, including noisy instances or outliers, and contribute minimally to centroid updates due to their low weights ($< 1$):

$$H_j^{(t)} = \left\{x_i \in C_j^{(t)} : s^{(t)}(x_i) > m_j^{(t)}\right\}, \text{ and } L_j^{(t)} = \left(C_j^{(t)} \setminus H_j^{(t)}\right). \qquad (6)$$

The distinction between core and peripheral points reflects the intrinsic quality of cluster assignments, with core points representing high-confidence regions.

### 4.2 THEORETICAL RESULTS

**Stability of High-Silhouette Regions $H_j^{(t)}$** Under a set of cluster regularity assumptions (see all assumptions in Appendix B.1), points in $H_j^{(t)}$ consistently maintain or improve their silhouette scores. Weighted centroid updates shift cluster centers toward these well-clustered regions, reducing intra-cluster distances without compromising inter-cluster separation. The slow drift of the median silhouette score (due to the weight dominance of high-confidence points) preserves membership in $H_j^{(t+1)}$. As a result, high-silhouette regions evolve smoothly and retain structural integrity (analysis is provided in Appendix B.2).

**Objective Function and Convergence** K-Sil is designed to optimize Silhouette (either $S_M$ or $S_m$), however, random drops in silhouette might occur across iterations, given its convex nature. Thus, we will use a modified version of Silhouette as the objective function that ensures convergence in a finite number of iterations and preserving the insights of silhouette metrics:

$$F = F^{(t)} = \sum_{j=1}^{k} \sum_{x_i \in C_j^{(t)}} w_i^{(t)} \cdot s^{(t)}(x_i). \tag{7}$$

This objective is bounded, as silhouette scores lie within $[-1, 1]$ (or $[0, 1]$ for non-trivial clusters), and total weight is fixed. It increases monotonically since updates prioritize core regions—high-weight points improve with each iteration, while low-weight points contribute little. Regularity in centroid movement and cluster separation preserves these gains. As $F^{(t)}$ is bounded and non-decreasing, and only finitely many clusterings exist, the algorithm converges in finite steps to a locally optimal partition (a detailed analysis of these properties is provided in Appendix B.3).

## 5 EMPIRICAL VALIDATION

To evaluate K-Sil we conducted experiments on both synthetic and real-world datasets, covering a range of clustering challenges such as varying densities, high-dimensional spaces, noisy structures and non-convex shapes. **Synthetic datasets** included: S1 (500 points distributed across 5 clusters of varying densities), S2 (500 points in 5 clusters of $\sigma = 1$ within a 12-dimensional space), S3 (similar to S2 with an increased cluster spread, $\sigma = 2.5$) and S4 (1500 points, combining a circular cluster, a line, and 50% noise). **Real-world datasets** comprised: Iris (150 samples of 4 numerical features, of 3 flower species), Mice Protein (1080 samples from 8 conditions, with 77 protein features), Glass (214 instances of 6 glass types, described by 9 elemental compositions), Wine (178 samples of 13 chemical attributes, from three wine types), Digits (1797 images of 64 pixel values, of handwritten digits 0-9) and a 20 Newsgroups Subset (2389 text documents from 4 overlapping categories). High-dimensional data were reduced via PCA (retaining $\geq 90\%$ of variance), text was TF-IDF vectorized and numerical features were standardized (all real-world datasets are publicly available from `UCI` or `scikit-learn`).

**Set-Up** To assess performance, we evaluate K-Sil against standard $k$-means, and two other instance-weighted variants; our own implementation of LOFK-Means Moggridge et al. (2020), and DensityKMeans, our instance-weighted $k$-means baseline, which estimates each point's local density as the average distance to its $h$ nearest neighbours and sets its weight to the inverse of this estimate. We designed DensityKMeans to better capture the inherent structure of datasets with varying densities by emphasizing points located in denser regions and reducing the contribution of those in sparser areas. All algorithms are initialized from the

same centroids to ensure a fair comparison. Additionally, for silhouette comparisons we tune hyperparameters via grid search to maximise the chosen internal objective ($S_M$ or $S_m$): neighbourhood size for LOFKMeans/DensityKMeans, and the weight-sensitivity $p$ and weighting scheme for K-Sil. Clustering performance is evaluated with Wilcoxon signed-rank tests on *paired* silhouette scores (K-Sil vs. each baseline). We sweep $k \in \{2, \ldots, 10\}$ with multiple trials per $k$; when the paired differences are significant ($p < 0.05$), we report the mean relative silhouette improvement across $k$. We then repeat the test at the ground-truth $k_{\mathrm{GT}}$. All tests are run separately for $S_M$ and $S_m$, matching the K-Sil configuration. As an additional check where internal and external structure coincide, we also report mean NMI (with 95% CIs) on synthetic datasets, where labels are ground truth by construction and align with the Euclidean cluster geometry and chosen $k$, showing that the internal gains translate to external agreement when objectives align.

Table 1: Mean relative macro-silhouette ($S_M$) gains (%) of K-Sil over baselines for $k = 2$–$10$; parentheses show values at $k_{\mathrm{GT}}$. (–) denotes non-significance (Wilcoxon, $p \geq 0.05$). See Appendix C for per-dataset settings, approximate-score comparisons, and $k$-wise trends.

| Dataset | Average Relative Improvements in $S_M$ of K-Sil | | |
| --- | --- | --- | --- |
| | over K-Means | over DensityK-Means | over LOFK-Means |
| S1 | 7.73% (2.55%) | 7.25% (4.95%) | 4.27% (1.29%) |
| S2 | 46.60% (42.56%) | 6.46% (-) | 5.63% (-) |
| S3 | 30.43% (47.32%) | 15.16% (-) | 6.70% (-) |
| S4 | 16.29% (28.22%) | 24.25% (24.50%) | 12.32% (27.35%) |
| Iris | 13.35% (1.66%) | 6.26% (-) | 9.22% (1.26%) |
| Mice Protein | 42.41% (45.80%) | 39.57% (48.60%) | 29.24% (34.29%) |
| Glass | 42.31% (50.25%) | 66.05% (62.83%) | 34.36% (36.55%) |
| Wine | 4.03% (-) | 5.10% (1.01%) | 3.02% (-) |
| Digits | 5.71% (9.02%) | 15.78% (23.43%) | 32.47% (28.08%) |
| 20 Newsgroups | 197.13% (254.05%) | - (-) | 212.65% (220.33%) |

Table 2: Mean relative micro-silhouette ($S_m$) gains (%) of K-Sil over baselines for $k = 2$–$10$; parentheses show values at $k_{\mathrm{GT}}$. (–) denotes non-significance (Wilcoxon, $p \geq 0.05$). See Appendix C for mean relative improvements on $S_m, S_M$ convex combinations.

| Dataset | Average Relative Improvements in $S_m$ of K-Sil | | |
| --- | --- | --- | --- |
| | over K-Means | over DensityK-Means | over LOFK-Means |
| S1 | 6.32% (0.46%) | 4.57% (2.26%) | - (-) |
| S2 | 36.00% (29.01%) | - (-) | - (-) |
| S3 | 18.86% (35.77%) | 6.04% (-) | - (-) |
| S4 | 7.22% (10.68%) | 17.98% (5.81%) | 4.03% (6.12%) |
| Iris | 8.56% (-) | 2.86% (-) | 4.55% (-) |
| Mice Protein | 13.24% (15.25%) | 11.02% (15.84%) | 8.02% (10.67%) |
| Glass | 23.20% (14.64%) | 54.29% (32.13%) | 17.84% (3.92%) |
| Wine | 2.49% (-) | 3.82% (-) | 1.40% (-) |
| Digits | 2.60% (4.78%) | 36.57% (24.23%) | 47.25% (21.38%) |
| 20 Newsgroups | 0.41% (4.37%) | 1327.89% (38.58%) | 25.86% (35.75%) |

**Empirical Results** Across identical initializations and a shared tuning protocol, K-Sil yields consistent gains over standard $k$-means and the instance-weighted baselines, with statistically significant improvements in both macro- and micro-averaged silhouette on most datasets and $k$ values (Table 1, 2). The largest gains occur in regimes that are challenging for centroid-based methods—overlap, noise, and imbalance—where silhouette-guided reweighting shifts centroids toward cohesive cores and away from boundary regions. Improvements hold not only at the ground-truth $k_{\mathrm{GT}}$ but also under misspecified $k$, indicating robustness

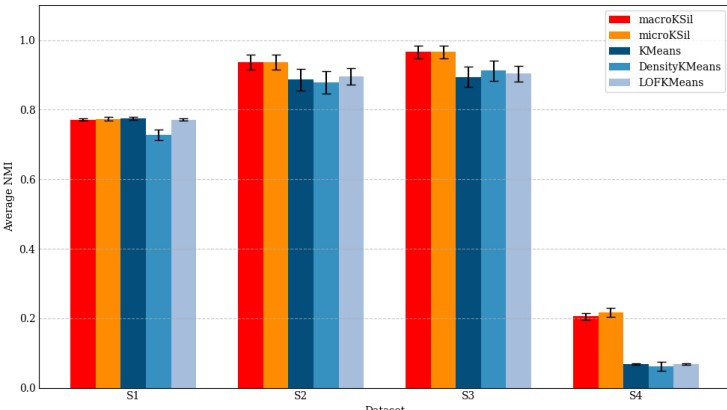

Figure 4: Average NMI scores with 95% (t-distribution) confidence intervals (error bars) of K-Sil (macro- and micro-silhouette modes), $k$-means, DensityKMeans, and LOFKMeans on synthetic datasets. LOFKMeans used 5 neighbours (`scikit-learn` default), and DensityKMeans used 10 neighbours for local density estimation, values appropriate for the size and structure of S1-S4.

to suboptimal parameter choices. Consistent with the design of K-Sil, gains are particularly pronounced for the macro objective $S_M$, whose emphasis on uniform per-cluster quality aligns with our cluster-centric weighting strategy.

Importantly, the sensitivity plots in Figure 3 include the point $p=0$, which *coincides with the best unweighted silhouette partition attained by the underlying k-means run*; K-Sil improves on this $p=0$ baseline over a broad range of $p > 0$, showing that the benefits are not due to retaining a favorable $k$-means iterate but to the silhouette-guided updates themselves. The same qualitative conclusions hold when silhouettes are approximated for scalability (Appendix Table 6). Finally, on synthetic datasets, where labels are ground truth by definition and align with Euclidean cluster geometry, average NMI with 95% confidence intervals (Figure 4) mirrors the internal gains, indicating that improved cohesion/separation translates to external agreement when objectives are aligned.

## 6 CONCLUSIONS

K-Sil enhances $k$-means by incorporating silhouette-driven instance weighting, amplifying high-confidence regions and suppressing noise within clusters. It supports user-defined objectives, macro-averaged silhouette for balanced cluster quality, micro-averaged for local cohesion, or combinations, while its cluster-centric weighting naturally aligns with macro-averaged silhouette's emphasis on structural balance. By addressing core limitations of $k$-means, K-Sil achieves statistically significant improvements over baselines across synthetic and real-world datasets, especially in noisy, imbalanced, or overlapping clusters. To improve efficiency without sacrificing accuracy, it employs objective-aware sampling and silhouette approximations that reduce computational overhead while preserving clustering quality. Nonetheless, K-Sil remains more computationally intensive than $k$-means due to its iterative computations. Its dependence on Euclidean distance also constrains applicability in non-Euclidean spaces, suggesting opportunities for extensions to more flexible distance metrics. While silhouette-based refinement improves clustering robustness, the algorithm still inherits sensitivity to initialization, making multiple restarts advisable. Overall, K-Sil offers a principled, flexible extension to $k$-means that excels in difficult clustering scenarios, with strong performance and interpretable behaviour. Future work may explore adaptive metrics and further optimizations.

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

# A  DETAILED METHODOLOGY

## A.1  K-SIL ALGORITHM OUTLINE

---

**Algorithm 1: K-Sil Clustering**

---

**Require:** Dataset $X$, number of clusters $k$, and parameters $P$ including:
- Centroid initialization method (random or $k$-means++),
- Chosen Silhouette objective $S$ ($S_M$, $S_m$, or a combination),
- Sampling size,
- Approximation option,
- Centroid movement threshold $\tau$ and Maximum number of iterations,
- Weighting scheme (power or exponential),
- Weight sensitivity parameter value [2]

**Ensure:** Final cluster centroids $\mu^*$, labels $L^*$ and objective score $S^*$
    $(\mu, L) \leftarrow$ Initial centroids and labels via one $k$-means iteration
    $S^* \leftarrow (-1)$ (best silhouette objective observed)
    $(\mu^*, L^*) \leftarrow (\mu, L)$ (centroids and labels corresponding to $S^*$)
    **repeat**
        **for** each cluster $C_j$ **do**
            **if** sampling is enabled **then**
                Let $C_j^S \subset C_j$ be the sampled subset
            **else**
                Let $C_j^S = C_j$
            **end if**
            Compute silhouette scores (or approximations) for all $x_i \in C_j^S$
            Assign weights for all $x_i \in C_j^S$ based on weighting scheme—sensitivity value
        **end for**
        $\mu \leftarrow$ Update cluster centroids using weighted averages
        $L \leftarrow$ Reassign each $x_i \in X$ to the nearest centroid
        Reinitialize any empty cluster centroids if necessary
        score $\leftarrow$ Compute current silhouette aggregation objective on new Labels $L$
        **if** score $> S^*$ **then**
            $S^* \leftarrow$ score
            $(\mu^*, L^*) \leftarrow$ Store current $\mu$ and $L$ as the best solution
        **end if**
    **until** (average centroid movement $< \tau$) or (maximum iterations reached)
    **return** $(\mu^*, L^*, S^*)$

---

## A.2  EVALUATION OF SILHOUETTE APPROXIMATION

To improve scalability in large datasets, K-Sil adopts a silhouette approximation (§ 3.1 equation 3) that avoids exact pairwise distance computations. Here, we evaluate this refined approximation, denoted as ApR, against a commonly used simplification, denoted as ApS, that estimates silhouette values based only on distances to centroids. Specifically, for a point $x_i \in C_j$, ApS defines $a(x_i)$ as the distance from $x_i$ to its assigned centroid $\mu_j$, and $b(x_i)$ as the distance to the nearest centroid of a different cluster, resulting to the silhouette approximation:

$$\tilde{s}(x_i) = \frac{\min_{h \neq j}\{\|x_i - \mu_h\|\} - \|x_i - \mu_j\|}{\max\{\|x_i - \mu_j\|,\ \min_{h \neq j}\{\|x_i - \mu_h\|\}\}}.$$

On the datasets used in § 5, using ground-truth $k$-means partitions, the refined approximation (ApR) consistently aligns better with exact silhouette scores, both at the point level and in aggregated metrics, than the simplified baseline (ApS).

---

[2]If auto-tuning for weight sensitivity is enabled, the algorithm is executed for each predefined candidate in a coarse, parallelized grid search, and the configuration yielding the highest silhouette objective is selected.

Table 3: Spearman Rank Correlation ($\rho$) of ApR and ApS point scores with exact point-silhouette scores. The highest correlation for each dataset is indicated in **bold**.

| Dataset | ApR Correlation ($\rho$) | ApS Correlation ($\rho$) |
|---|---|---|
| S1 | **0.994** | 0.958 |
| S2 | **1.000** | 0.935 |
| S3 | **0.999** | 0.985 |
| S4 | **0.965** | 0.919 |
| Iris | **0.984** | 0.906 |
| Mice Protein | **0.997** | 0.846 |
| Glass | **0.988** | 0.917 |
| Wine | **0.984** | 0.915 |
| Digits | **0.998** | 0.961 |
| 20 Newsgroups | **0.979** | 0.809 |

Table 4: Comparison of Exact, ApR and ApS aggregated scores.

| Dataset | Exact (Silhouette) Scores | | ApR Scores | | ApS Scores | |
|---|---|---|---|---|---|---|
| | $S_{\mathbf{m}}$ | $S_{\mathbf{M}}$ | $S_{\mathbf{m}}$ | $S_{\mathbf{M}}$ | $S_{\mathbf{m}}$ | $S_{\mathbf{M}}$ |
| S1 | 0.576 | 0.565 | **0.543** | **0.532** | 0.680 | 0.671 |
| S2 | 0.813 | 0.813 | **0.811** | **0.811** | 0.868 | 0.868 |
| S3 | 0.405 | 0.349 | **0.386** | **0.338** | 0.514 | 0.444 |
| S4 | 0.306 | 0.308 | **0.264** | **0.271** | 0.456 | 0.456 |
| Iris | 0.479 | 0.443 | **0.440** | **0.394** | 0.611 | 0.582 |
| Mice Protein | 0.133 | 0.166 | **0.126** | **0.156** | 0.231 | 0.272 |
| Glass | 0.330 | 0.378 | **0.287** | **0.336** | 0.494 | 0.541 |
| Wine | 0.568 | 0.552 | **0.523** | **0.502** | 0.671 | 0.661 |
| Digits | 0.194 | 0.198 | **0.183** | **0.186** | 0.296 | 0.301 |
| 20 Newsgroups | 0.066 | 0.086 | **0.056** | **0.082** | 0.129 | 0.143 |

### A.3 EMPTY-CLUSTER RE-INITIALIZATION STRATEGY

If a cluster $C_j^{(t)}$ becomes empty during point reassignment after the centroid update, it is re-initialized by selecting the point farthest from the centroid of the largest cluster (by size), denoted $C_{\max}^{(t)}$. Formally, the re-initialized centroid would be:

$$\mu_j^{(t+1)} = \arg \max_{x \in C_{\max}^{(t)}} \|x - \mu_{\max}^{(t)}\|.$$

This strategy is based on the intuition that points farthest from the dominant cluster's center are likely outliers or represent under-clustered substructures. Reassigning such a point as the new centroid helps restore the expected number of clusters while potentially revealing overlooked data regions.

An alternative approach could involve choosing the cluster with the highest variance instead of the largest size, under the assumption that higher spread may indicate unresolved internal structure:

$$\mu_j^{(t+1)} = \arg \max_{x \in C_{\mathrm{var}}^{(t)}} \|x - \mu_{\mathrm{var}}^{(t)}\|, \quad \text{where } C_{\mathrm{var}} = \arg \max_{C_h} \frac{1}{|C_h^{(t)}|} \sum_{x \in C_h} \|x - \mu_h^{(t)}\|^2.$$

However, size-based selection is simpler, less sensitive to noise, and tends to be more stable in practice, particularly when outliers dominate high-variance clusters.

### A.4 HIGHEST SILHOUETTE ($S^*$) PARTITION

As demonstrated in Analysis (§4), K-Sil's weighted-SSE centroid updates increase the modified weighted-silhouette objective $F$ (equation 7) until conver-

gence. However, because instance weights $w_i^{(t)}$ adapt at every iteration, there can be scenarios where the final iterations further refine the modified weighted objective $F^{(t)}$ by adjusting centroids toward highly-weighted regions. This refinement can narrow cluster separations and consequently reduce the unweighted macro- or micro-averaged silhouette scores, resulting in a correlation divergence between $F$ and $S_M$ or $S_m$ in the final updates of the algorithm. In such cases, the unweighted silhouette may reach its maximum at an earlier iteration prior to convergence.

Recording the unweighted silhouette score $S^{(t)}$ at each step and retaining the labels corresponding to its maximum $S^*$ effectively mitigates the risk of overfitting to the weighted objective $F$, ensuring that the final output remains faithful to the original silhouette-based clustering goal. Additionally, even when the retained clustering corresponds to an intermediate iteration rather than the final one, it is not an unstable or noisy partition. Since low-silhouette points contribute negligibly to centroid updates due to their small weights, they cannot falsely drive a silhouette spike. The inherent robustness of the silhouette-based instance-weighting ensures that high-silhouette partitions reflect genuinely stable and well-separated structures, not effects of noise or short-lived configurations.

## B  DETAILED ANALYSIS

### B.1  REGULARITY ASSUMPTIONS & ALGORITHMIC PROPERTIES

Our analysis relies on a set of assumptions and properties which, while reflecting idealized conditions, offer a clear framework of the algorithm's behavior.

1. At each iteration $t$, clusters $C_j^{(t)}$ form convex sets, and well-clustered points $H_j^{(t)}$ lie within the convex hull of their points (concentrated near the centroid), ensuring centroids evolve within well-defined, dense regions, avoiding erratic shifts.

2. The total weight of high-silhouette score points exceeds that of low-silhouette ones: $\sum_{x_i \in H_j^{(t)}} w_i \geq \gamma \cdot \sum_{x_i \in C_j^{(t)}} w_i$, with $\gamma > 0.5$. This arises naturally from the weighting schemes, which prioritize points above the median silhouette score.

3. Centroid movements $\Delta\mu_j^{(t)} = (\mu_j^{(t+1)} - \mu_j^{(t)})$ align directionally with well-clustered points: $\Delta\mu_j^{(t)} \cdot (x_i - \mu_j^{(t)}) \geq 0 \ \forall x_i \in H_j^{(t)}, \ j \in \{1, \ldots, k\}$, reducing intra-cluster distances for these points and preserving compactness. A property that is a direct outcome of the weighted mean update, which pulls centroids toward core regions.

4. For every pair of distinct clusters $C_j^{(t)}$, $C_h^{(t)}$ with $j \neq h$, centroid updates do not systematically reduce inter-centroid distances: $\|\mu_j^{(t+1)} - \mu_h^{(t+1)}\| \geq \|\mu_j^{(t)} - \mu_h^{(t)}\|$ (based on weighting schemes, the centroids move toward their respective high-silhouette regions, not toward other clusters).

5. Silhouette scores change smoothly with centroid positions—small centroid movements induce bounded changes in silhouette scores. A Lipschitz constant $L$ quantifies this relationship: $|s^{(t+1)}(x_i) - s^{(t)}(x_i)| \leq L\|\mu_j^{(t+1)} - \mu_j^{(t)}\|$, where $s^{(t)}(x_i)$ represents the silhouette score of a point $x_i \in C_j^{(t)}$ at iteration $t$.

### B.2  STABILITY OF HIGH-SILHOUETTE REGIONS $H_j^{(T)}$

**Non-Decreasing Silhouette Scores for High-Silhouette Score Points**  *For any high-silhouette score point, $x_i \in H_j^{(t)}$ $(j = 1, \ldots, k)$, its silhouette score does not decrease after the centroid update:*

$$x_i \in H_j^{(t)} \Rightarrow s^{(t+1)}(x_i) \geq s^{(t)}(x_i).$$

By assumption (1), each cluster $C_j^{(t)}$ is convex and by assumption (2), the centroid update is primarily influenced by points in the well clustered region $H_j^{(t)}$, thus the

updated centroid $\mu_j^{(t+1)}$ moves closer to or within the convex hull formed by high-silhouette score points.

Due to centroid shift alignment (3), the centroid movement is directed towards these high-silhouette score points, decreasing or maintaining intra-cluster distances for them. Thus, for any $x_i \in H_j^{(t)}$:

$$\|x_i - \mu_j^{(t+1)}\| \leq \|x_i - \mu_j^{(t)}\| \Rightarrow \alpha^{(t+1)}(x_i) \leq \alpha^{(t)}(x_i).$$

Additionally, assumption (4) ensures clusters' centroids do not move closer to each other, implying inter-cluster distances remain stable or increase, thus:

$$b^{(t+1)}(x_i) \geq b^{(t)}(x_i).$$

Combining these two inequalities directly gives:

$$s^{(t+1)}(x_i) \geq s^{(t)}(x_i) \ \ \forall x_i \in H_j^{(t)} \text{ for any } j \in \{1, \ldots, k\}.$$

**Stability of High-Silhouette Score Points** *For any well-clustered point, $x_i \in H_j^{(t)}$ ($j = 1, \ldots, k$), it remains in the high-silhouette region after the centroid update:*

$$x_i \in H_j^{(t)} \Rightarrow x_i \in H_j^{(t+1)}$$

From assumption (5) (Lipschitz continuity of silhouette scores), the median silhouette scores between consecutive iterations satisfy:

$$|m_j^{(t+1)} - m_j^{(t)}| \leq L \cdot \|\mu_j^{(t+1)} - \mu_j^{(t)}\|.$$

We denote total weights as:

$$W_H = \sum_{x_i \in H_j^{(t)}} w_i, \ \ W = \sum_{x_i \in C_j^{(t)}} w_i.$$

And by assumption (2), we have the inequality: $W_H \geq \gamma \cdot W \Rightarrow \frac{W_H}{W} \geq \gamma > 0.5$. By centroid update definition, we have:

$$\mu_j^{(t+1)} = \frac{1}{W} \sum_{x_i \in C_j^{(t)}} w_i x_i \Rightarrow \left(\mu_j^{(t+1)} - \mu_j^{(t)}\right) = \frac{1}{W} \sum_{x_i \in C_j^{(t)}} w_i(x_i - \mu_j^{(t)})$$

$$\Rightarrow \|\mu_j^{(t+1)} - \mu_j^{(t)}\| = \left\|\frac{1}{W} \sum_{x_i \in C_j^{(t)}} w_i(x_i - \mu_j^{(t)})\right\| \leq \frac{1}{W} \sum_{x_i \in C_j^{(t)}} w_i\|x_i - \mu_j^{(t)}\|.$$

And since $(H_j^{(t)} \cup L_j^{(t)}) = C_j^{(t)}$ with $(H_j^{(t)} \cap L_j^{(t)}) = \emptyset$, we have:

$$\|\mu_j^{(t+1)} - \mu_j^{(t)}\| \leq \frac{1}{W} \left( \sum_{x_i \in H_j^{(t)}} w_i\|x_i - \mu_j^{(t)}\| + \sum_{x_i \in L_j^{(t)}} w_i\|x_i - \mu_j^{(t)}\| \right), \text{ where:}$$

$$\sum_{x_i \in H_j^{(t)}} w_i\|x_i - \mu_j^{(t)}\| \leq \max_{x_i \in H_j^{(t)}} \|x_i - \mu_j^{(t)}\| \sum_{x_i \in H_j^{(t)}} w_i = \max_{x_i \in H_j^{(t)}} \|x_i - \mu_j^{(t)}\| \cdot W_H$$

and similarly $\sum_{x_i \in L_j^{(t)}} w_i\|x_i - \mu_j^{(t)}\| \leq \max_{x_i \in L_j^{(t)}} \|x_i - \mu_j^{(t)}\| \cdot (W - W_H)$.

Since by assumption (1), the centroid $\mu_j^{(t)}$ lies in the convex hull formed of $H_j^{(t)}$, high-silhouette score points cannot be farther from the centroid than half the cluster diameter: $\max_{x_i \in H_j^{(t)}} \|x_i - \mu_j^{(t)}\| \leq \frac{\text{diam}(C_j^{(t)})}{2}$, while low-silhouette score points

are at most at the full diameter distance: $\max_{x_i \in L_j^{(t)}} \|x_i - \mu_j^{(t)}\| \leq \text{diam}(C_j^{(t)})$. Thus we have:

$$\|\mu_j^{(t+1)} - \mu_j^{(t)}\| \leq \frac{1}{W} \left( \frac{W_H \cdot \text{diam}(C_j^{(t)})}{2} + (W - W_H) \cdot \text{diam}(C_j^{(t)}) \right)$$

$$\Rightarrow \|\mu_j^{(t+1)} - \mu_j^{(t)}\| \leq \left(1 - \frac{W_H}{2W}\right) \cdot \text{diam}(C_j^{(t)}) \Rightarrow \|\mu_j^{(t+1)} - \mu_j^{(t)}\| \leq \left(1 - \frac{\gamma}{2}\right) \cdot \text{diam}(C_j^{(t)})$$

Therefore, the median silhouette score shift in cluster $C_j^{(t)}$ is bounded by:

$$|m_j^{(t+1)} - m_j^{(t)}| \leq L \cdot \left(1 - \frac{\gamma}{2}\right) \cdot \text{diam}(C_j^{(t)}) = \epsilon(\gamma).$$

For a high-silhouette score point, $x_i \in H_j^{(t)}$, we know that $s^{(t+1)}(x_i) \geq s^{(t)}(x_i)$ and by definition we have that $s^{(t)}(x_i) > m_j^{(t)}$ or equivalently there exists a $\delta_i \geq 0$ such that $s^{(t)}(x_i) - m_j^{(t)} > \delta_i$. Thus, we get: $s^{(t+1)}(x_i) - m_j^{(t)} > \delta_i \geq 0$.

Since $\epsilon(\gamma)$ is a continuous, strictly decreasing function of $\gamma$, by the Intermediate Value Theorem there exists a $\gamma^*$ such that $\epsilon(\gamma^*) = \delta_i$. By choosing $\gamma$ sufficiently large (by emphasizing points with high silhouette scores sufficiently strongly, while de-emphasizing low silhouette scores points) such that $\gamma > \gamma^*$ we ensure that $\epsilon(\gamma) < \delta_i$ and hence $s^{(t+1)}(x_i) - m_j^{(t)} > \delta_i > \epsilon(\gamma) \Rightarrow s^{(t+1)}(x_i) > m_j^{(t)} + \epsilon(\gamma) > m_j^{(t+1)}$. Therefore, by definition, $x_i \in H_j^{(t+1)}$, ensuring stability of high-silhouette regions across iterations.

### B.3 CONVERGENCE

To simplify the analysis, we assume steady weights for high-silhouette and low-silhouette points by considering the mean weight per cluster's subset ( 4.1). Specifically, we define:

$$w_{\text{high},j} = \frac{1}{|H_j^{(t)}|} \sum_{x_i \in H_j^{(t)}} w_i^{(t)}, \ w_{\text{low},j} = \frac{1}{|L_j^{(t)}|} \sum_{x_i \in L_j^{(t)}} w_i^{(t)},$$

in this way we smooth the weight variations and we maintain the weight-mechanism: $w_{\text{high},j} > 1 > w_{\text{low},j} \geq 0$.

**Boundedness** The objective function is bounded by the total weight of points, as silhouette scores are inherently limited to $[-1, 1]$:

$$F = \sum_{j=1}^{k} \sum_{x_i \in C_j^{(t)}} w_i^{(t)} \cdot s(x_i) \leq \sum_{j=1}^{k} \sum_{x_i \in C_j^{(t)}} w_i^{(t)} \cdot 1 = \sum_{x_i \in X} w_i^{(t)} = W \ \text{ (upper bound)}.$$

**Monotonic Improvement** We will evaluate every possible case of point movement between two iterations and determine its impact on the objective function. The net contribution of any point $x_i$ to the objective function $F^{(t)}$ as it transitions across iterations $(t \to t+1)$ is given by:

$$\Delta F_i^{(t)} = w_i^{(t+1)} s^{(t+1)}(x_i) - w_i^{(t)} s^{(t)}(x_i),$$

where $w_i^{(t+1)}$ and $w_i^{(t)}$ are the weights assigned to $x_i$ in iterations $t+1$ and $t$ based on its silhouette scores $s^{(t+1)}(x_i)$ and $s^{(t)}(x_i)$.

By the stability of high-silhouette regions B.2 we know that points in $H^{(t)}$ maintain or increase their silhouette scores and cannot transition to $L^{(t+1)}$. Since they retain their high weight, it follows that their contribution to the objective function is non-negative:

$$\Delta F_i^{(t)} = w_{\text{high},j} \left( s^{(t+1)}(x_i) - s^{(t)}(x_i) \right) \geq 0 \ (j = 1, \ldots, k).$$

For points in $L_j^{(t)}$ (for any cluster $C_j$) that remain borderline, their weights remain $w_{\text{low},j}$ and although their silhouette score might decrease, there will be a negligible effect on the objective function due to their low weight, meaning $\Delta F_i^{(t)} \approx 0$.

Additionally, when a low-silhouette score point, $x_i \in B_j^{(t)}$, transitions to another cluster ($C_l$, either in $H_l$ or $L_l$), its reassignment is based on reducing intra-cluster distance. By cluster separation assumption (4), centroid updates maintain or improve separation, ensuring an increase in its silhouette score and leading to $\Delta F_i^{(t)} \geq 0$.

Lastly, if the weighting scheme allows a point in $B_j^{(t)}$ to transition to $H_j^{(t+1)}$, it undergoes both an increase in weight and in silhouette score, ensuring

$$\Delta F_i^{(t)} = w_{\text{high},j} s^{(t+1)}(x_i) - w_{\text{low},j} s^{(t)}(x_i) > 0.$$

Summing over all points (cases), we conclude:

$$\Delta F^{(t)} = \sum_{x_i \in X} \Delta F_i^{(t)} \geq 0 \ \ \forall \text{ iteration } t.$$

Given that, the objective function is bounded above, non-decreasing, and the number of distinct partitions of $n$ points to $k < n$ clusters is finite ($k^n$), it follows that $F^{(t)}$ stabilizes in finite number of iterations. Therefore, K-Sil algorithm terminates at a locally optimal clustering configurations in finite time.

## C  EXTENDED EMPIRICAL RESULTS

Table 5: Weighting schemes used in K-Sil for macro-averaged silhouette score statistical comparison against baselines. "Across $k$" refers to the configuration used for the overall comparison across $k \in \{2, \ldots, 10\}$, and "At $k_{GT}$" refers to the configuration used only at the ground-truth number of clusters.

| Dataset | Weighting Schemes Used for $S_M$ Comparison | |
| --- | --- | --- |
| | **Across $k$** | **At $k_{GT}$** |
| S1 | Power | Power |
| S2 | Power | Exponential |
| S3 | Power | Exponential |
| S4 | Exponential | Exponential |
| Iris | Exponential | Exponential |
| Mice Protein | Exponential | Power |
| Glass | Exponential | Exponential |
| Wine | Exponential | Exponential |
| Digits | Power | Power |
| 20 Newsgroups | Power | Power |

Table 6: Relative improvements (%) in macro-averaged silhouette score of K-Sil (using approximate silhouette scores A.2, prioritizing $S_M$ - same configuration as in table 5) over $k$-means. Results are statistically significant ($p < 0.05$) across $k \in \{2, \ldots, 10\}$ and at the ground-truth number of clusters $k_{GT}$ ("-" for insignificant improvements).

| Dataset | Mean Relative Improvement ($S_M$) | |
|---|---|---|
| | Across $k$ (weighting scheme) | On $k_{GT}$ (weighting scheme) |
| S1 | 10.03% (Power) | 4.29% (Power) |
| S2 | 40.18% (Power) | 71.22% (Exponential) |
| S3 | 31.95% (Power) | 72.85% (Exponential) |
| S4 | 16.83% (Exponential) | 23.90% (Exponential) |
| Iris | 19.13% (Exponential) | 1.58% (Exponential) |
| Mice Protein | 79.95% (Exponential) | 74.35% (Power) |
| Glass | 48.95% (Exponential) | 40.02% (Exponential) |
| Wine | 4.55% (Exponential) | - |
| Digits | 3.77% (Power) | 9.46% (Power) |
| 20 Newsgroups | 183.57% (Power) | 237.74% (Power) |

Table 7: Statistical comparison of silhouette score performance between K-Sil (prioritizing a convex combination of micro and macro-silhouette: $0.5S_m + 0.5S_M$) and $k$-means. We report the average relative improvements (%) in $S_m$ and $S_M$ (along with the weighting scheme used, P: Power or E: Exponential) of K-Sil over $k$-means, across $k \in \{2, \ldots, 10\}$ and at the ground-truth number of clusters $k_{GT}$ for the statistically significant ($p < 0.05$) results for each dataset ("-"for statistically insignificant improvements).

| Dataset | Mean Relative Improvement on | | | |
|---|---|---|---|---|
| | ($S_M$) | | ($S_m$) | |
| | Across $k$ | On $k_{GT}$ | Across $k$ | On $k_{GT}$ |
| S1 | 7.58% (P) | 2.49% (P) | 6.19% (P) | - |
| S2 | 46.51% (P) | 42.55% (E) | 35.71% (P) | 29.01% (E) |
| S3 | 30.30% (P) | 47.32% (E) | 18.52% (P) | 35.77% (E) |
| S4 | 17.21% (E) | 29.01% (E) | 5.34% (P) | 8.63% (E) |
| Iris | 13.17% (P) | 1.60% (E) | 7.78% (P) | - |
| Mice Protein | 42.52% (E) | 45.35% (P) | 10.14% (P) | 10.23% (P) |
| Glass | 42.80% (E) | 49.35% (E) | 22.08% (P) | 13.71% (P) |
| Wine | 3.76% (E) | - | 2.46% (E) | - |
| Digits | 5.54% (P) | 9.00% (P) | 2.21% (P) | 4.65% (P) |
| 20 Newsgroups | 168.02% (E) | 241.92% (P) | - | - |

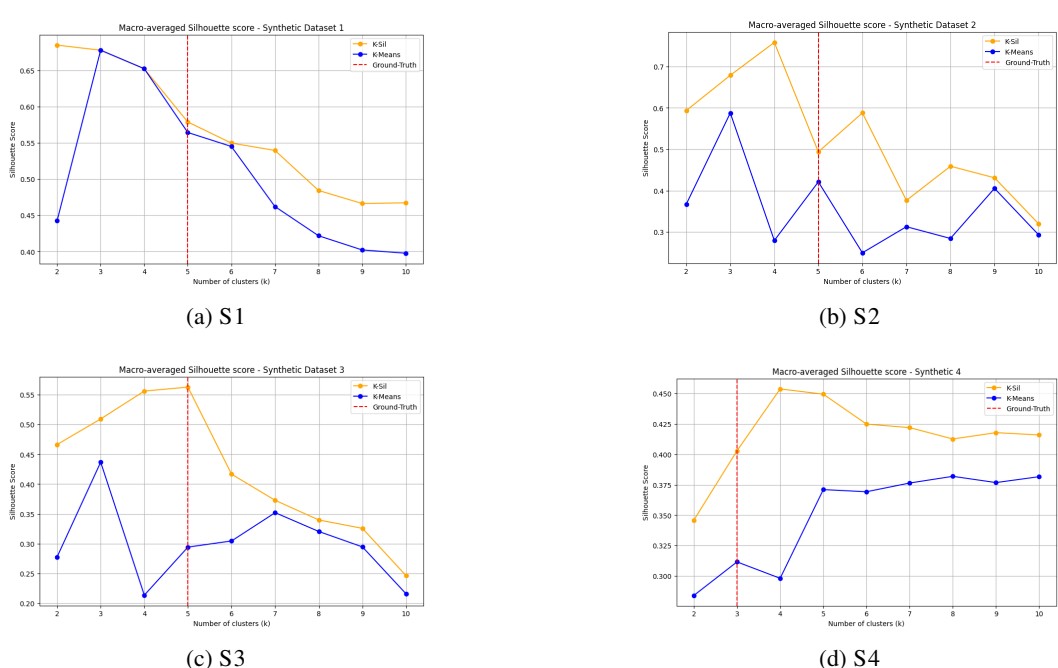

(a) S1

(b) S2

(c) S3

(d) S4

Figure 5: $S_M$ of K-Sil (orange) and standard $k$-means (blue) across the number of clusters $k \in \{2, \ldots, 10\}$ for synthetic datasets (the ground-truth number of clusters $k_{GT}$ is indicated with a red dashed line). Both algorithms share the same initial centroids for each run.

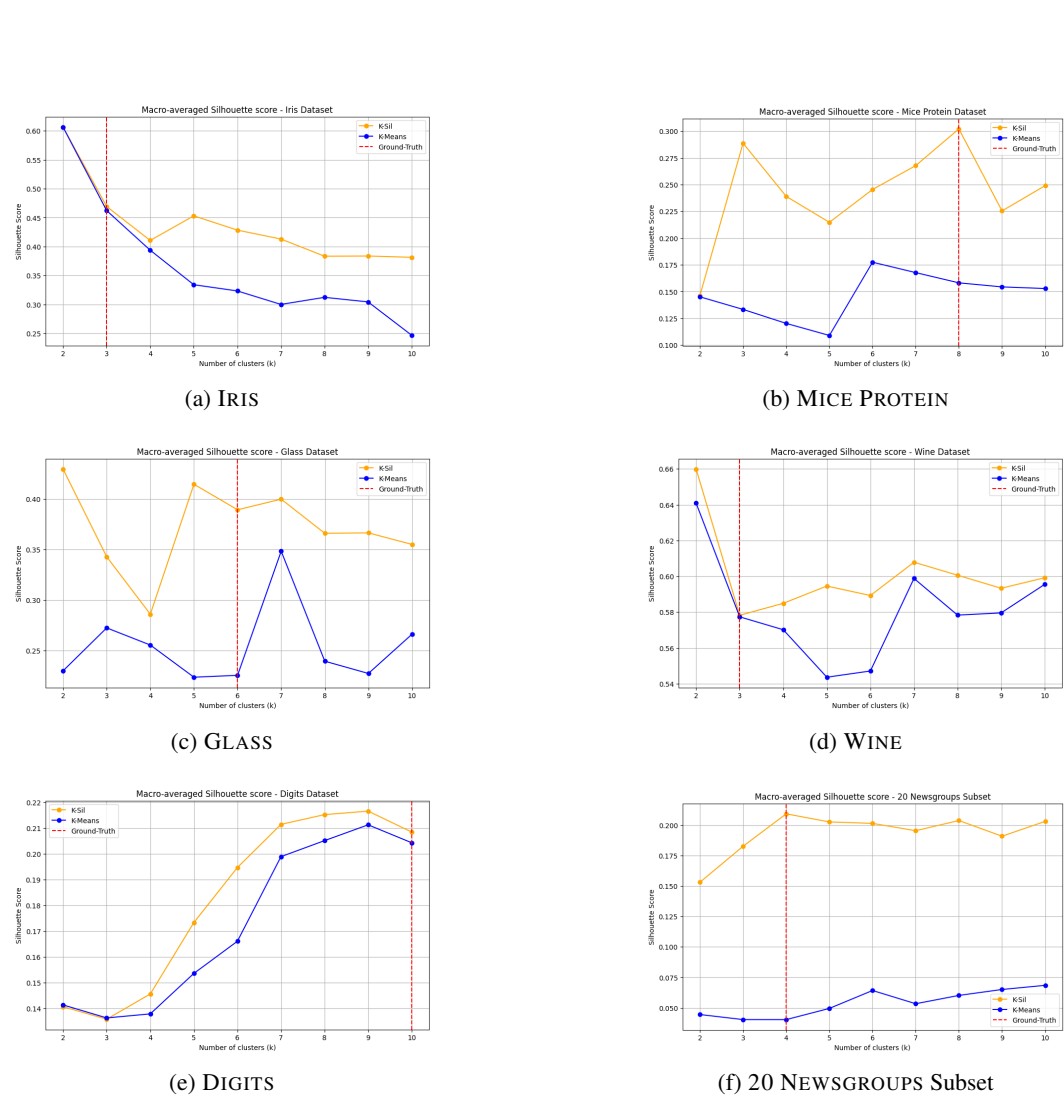

(a) IRIS

(b) MICE PROTEIN

(c) GLASS

(d) WINE

(e) DIGITS

(f) 20 NEWSGROUPS Subset

Figure 6: Macro-averaged silhouette scores for K-Sil (orange) and standard $k$-means (blue) across the number of clusters $k \in \{2, \dots, 10\}$ for real-world datasets (the ground-truth number of clusters $k_{GT}$ is indicated with a red dashed line).

Both algorithms share the same initial centroids for each run. K-Sil is configured to prioritze $S_{\mathrm{M}}$, utilizing the optimal weighting scheme with auto-tuned weight-sensitivity parameter for each $k$.

