# OpenReview forum: "K-Sil: Silhouette-Driven Instance-Weighted Clustering"
_ICLR.cc/2026/Conference — ICLR 2026 Conference Withdrawn Submission_

### Official Review · Reviewer_JNnN · 2025-10-26

**Soundness:** 1
**Presentation:** 2
**Contribution:** 2
**Rating:** 2
**Confidence:** 4

**Summary:**

The paper proposes a refinement of the k-means by proposing a novel optimization problem and provides some empirical analysis.

While the proposed problem is interesting and **may** indeed outperform the state-of-the-art methods, for me, there are not enough analytical results (specifically, approximation guarantees for the solution of the optimization problem) to give it "accept" as a theory paper, and not enough empirical evaluation to give it "accept" as a practical paper.

Given this, my assessment is a **reject**, especially since I believe that a more thorough analysis or empirical evaluation can elevate this paper significantly.

I found the core idea interesting and well motivated, and with stronger analysis or broader experiments, I believe this work **could** become a solid contribution. I look forward to seeing a more developed version in the future or during the rebuttal phase.

**Strengths:**

The problem is well-motivated, practical, and the proposed method indeed outperforms the comparison methods used.

The proposed problem is indeed original (at least to my knowledge), and the writing is mostly clear and of high quality.

I also appreciate the proof of convergence of the proposed methods.

The code being available (anonymously) was a good choice that I applaud the authors for (since, to my assessment, some reviewers may be overly nitpicky on the code), and allowed me to more clearly assess some aspects (specifically how the problem was solved).

**Weaknesses:**

The theoretical results seem to be only "STABILITY OF HIGH-SILHOUETTE REGIONS" and "CONVERGENCE", while showing convergence is good, to accept the paper as a theory paper, I want approximation guarantees.

The theoretical results could be rewritten slightly for formality; see suggestions.

There are only 3 comparison methods, 1 is k-means (which has the same initialization as the proposed method), and the other two are implemented by the authors, with one proposed by the authors.

For a practical paper, a significantly more comprehensive evaluation is needed, including comparisons with several of the methods cited in the related work or others. At a minimum, I would expect comparisons against major clustering methods from widely-used libraries (e.g., scikit-learn) that allow explicit control of the number of clusters: K-Means (already included), Spectral Clustering, Ward Hierarchical Clustering, Agglomerative Clustering, and Bisecting K-Means. Additionally, the evaluation should include at least three strong baselines from prior works referenced in the paper.

All the compression methods, besides k-means, are implemented by the authors, which weakens the empirical credibility. While implementing some methods is understandable, in this case, it seems excessive, given the low count of competing methods considered.

For the empirical comparison, I would appreciate more imbalanced datasets; see suggestion 3 for a suggested dataset.

**Questions:**

The writing is rather clear, and the proposed problem and its proposed solution are understandable.
As such, I have no significant questions of note.

# Suggestions
1. While minor, in the theoretical part, use explicit lemmas, theorems, claims, assumptions, etc. For example, in section B.2. "Non-Decreasing Silhouette Scores for High-Silhouette Score Points" should be a lemma/theorem followed by a proof, and currently it reads as a paragraph.
2. Either:
- Add more comprehensive empirical evaluations, using some of the many works cited in the related work, or other recent works.
- Provide approximation guarantees for the suggested method, so it could fall under a theory paper, and as such, the evaluation could be more relaxed (still a few additional methods seem needed).
3. Add a synthetic dataset considering imbalanced clusters, for example, the following dataset on the plane consisting of two clusters.
- A uniform sample of 1000 points from the unit circle.
- A uniform sample of 25 points, where each point is chosen from a disc of radius 0.1, centered at (2, 0).
4. It seems that the running time was not provided in the empirical section. It is important, given the changes mentioned in the paper, to this end, and for a more comprehensive comparison. Also, given that the running time of the solution is only shown to be finite (and not provided with an explicit bound), I believe that the runtime for at least one test should be plotted as a histogram. This would allow a rough understanding of the runtime distribution, for example, whether the tail decays rapidly (and can thus be empirically ignored) or decays slowly, making the expected runtime heavily affected by the not-so-rare, long-running cases.
5. While minor, the caption of the algorithm seems informal. I would suggest K-Sil-Clustering(X,...) over the current "K-Sil Clustering", which doesn't name the parameters explicitly.
6. The algorithm should be formalized. For example, instead of just "Sampling size" and then "Let $C_j^S \subset C_j$ be the sampled subset", explicitly state *$\lambda\geq 1$ sample size* and then *Let $C_j^S \subset C_j$ be a uniform i.i.d. sample of size $\lambda$*, or something along the lines.
7. While purely stylistic, you might consider using algorithm2e for a cleaner layout (vertical scope lines), and the Input/Output convention instead of Require/Ensure for clarity.

---

### Official Review · Reviewer_DeGn · 2025-10-29

**Soundness:** 2
**Presentation:** 2
**Contribution:** 2
**Rating:** 4
**Confidence:** 5

**Summary:**

This paper addresses the limitations of traditional k-means in handling outliers, imbalanced clusters, and noisy data by proposing K-Sil, a silhouette-guided instance-weighted clustering algorithm. The work demonstrates clear innovation in integrating internal validation metrics into the clustering process and provides rigorous theoretical and empirical support.

**Strengths:**

The proposed silhouette-driven instance-weighting mechanism represents a meaningful advance over traditional k-means and existing weighted variants. By prioritizing high-confidence points and suppressing noisy or borderline regions, the algorithm effectively mitigates negative clustering outcomes while maintaining computational feasibility.
The authors clearly identify core problems in clustering—centroid skewing due to outliers and poor separation in imbalanced clusters—and address the critical issue of negative knowledge transfer.
The experimental design is rigorous, covering multiple synthetic and real-world datasets and comparing against relevant baselines.
The inclusion of scalability strategies, such as silhouette approximation and objective-aware sampling, makes the method applicable to large-scale datasets

**Weaknesses:**

While the paper claims scalability via sampling and approximation, it does not provide quantitative comparisons of runtime/memory usage between K-Sil and baselines (e.g., how much longer K-Sil takes than k-means on large datasets like 20 Newsgroups). This makes it hard to assess whether K-Sil’s performance gains justify its additional computational cost, especially for time-sensitive applications.
Like standard k-means, K-Sil inherits sensitivity to centroid initialization (the paper uses k-means++ or random initialization but does not analyze initialization impact). The authors mention “multiple restarts are advisable” in the conclusion but provide no empirical evidence: e.g., how often K-Sil falls into poor local optima compared to baselines, or whether initialization strategies like k-means++ consistently outperform random seeding across datasets.
Highly imbalanced clusters: For example, datasets with one cluster containing 90% of points and others with 1–2%—does
SM-guided K-Sil still maintain small clusters’ integrity, or does it prioritize the dominant cluster?

**Questions:**

To enhance the work further, we recommend:
Extending K-Sil to non-Euclidean metrics and providing compatibility analysis for the silhouette approximation.
Adding empirical studies on initialization sensitivity and comparing empty-cluster reinitialization strategies.
Quantifying computational overhead and clarifying hyperparameter tuning protocols.
Testing edge cases (extreme imbalance, high noise) to validate robustness.

---

### Official Review · Reviewer_6UiW · 2025-10-31

**Soundness:** 2
**Presentation:** 1
**Contribution:** 2
**Rating:** 4
**Confidence:** 5

**Summary:**

This paper proposes K-Sil, a clustering algorithm that introduces a silhouette-driven instance weighting mechanism into the k-means framework. Instead of using uniform sample weights, K-Sil iteratively reweights each instance based on its (approximate) silhouette value, using either a Power or Exponential weighting scheme. The algorithm incorporates centroid updates with weighted means and a convergence mechanism that records the best unweighted silhouette partition. To reduce computational cost, the paper derives an efficient approximation of the silhouette components $\tilde{a}(x_i)$ and $\tilde{b}(x_i)$, lowering the complexity from $O(n^2)$ to roughly $O(nd)$. Experiments across several datasets show consistent improvements over K-means and other baseline methods in terms of both macro- and micro-average silhouette scores.

**Strengths:**

1. This paper propose a more effective reduction of silhouette computation cost via the approximation method.

2. Empirical results demonstrate consistent improvements over standard k-means and some variants on macro-silhouette、micro-silhouette and NMI.

3. The proposed Weighting Scheme (Power and Exponential weighting schemes), is designed for different data characteristics to obtain different weights for updating suitable centroids.

**Weaknesses:**

1. The experimental validation of the silhouette approximation is not sufficiently persuasive. The largest dataset used is 20 Newsgroups with only 2,389 samples, which is relatively small. Larger and more diverse datasets (e.g., MNIST, KDD Cup 1999, or HIGGS) should be included to better demonstrate the scalability and effectiveness of the proposed silhouette approximation.

2. Table 4 compares the aggregated scores of Exact, ApR, and ApS methods, but the corresponding runtime analysis is missing. Runtime comparisons between the proposed methods and other competitors should be reported to substantiate the claimed computational efficiency.

3. The experimental section focuses primarily on silhouette-based metrics and NMI. Including additional external validation measures (e.g., clustering accuracy or ARI) would make the empirical claims more convincing.

4. The sensitivity of the algorithm to hyperparameters such as pand τis only discussed qualitatively. A more systematic quantitative analysis would strengthen the experimental rigor.

**Questions:**

1. Does the algorithm ever oscillate between cluster configurations due to the dynamic reweighting process? How frequently is the “best unweighted silhouette partition” selected as the final output in practice?

2. When approximate silhouettes are used, to what extent can the deviation from the true silhouette increase before the clustering quality degrades noticeably?

3. Do the clustering accuracy and within-cluster cost ($\sum_{x_i \in C_j} \sum \|x_i - \mu_j\|^2$) correlate positively with the silhouette score? The authors could provide additional experiments to verify this relationship.

---

### Official Review · Reviewer_6CB8 · 2025-10-31

**Soundness:** 3
**Presentation:** 3
**Contribution:** 2
**Rating:** 4
**Confidence:** 4

**Summary:**

The proposed method (K-Sil) extends k-means by incorporating silhouette-based instance weighting, using each point’s silhouette score to emphasize well-clustered regions and down-weight noisy or ambiguous points. It introduces an approximation to the silhouette score that replaces costly pairwise distance computations with centroid- and dispersion-based estimates, reducing the complexity from O(n^2) to roughly O(nk). This approximation offers a trade-off between computational efficiency and the fidelity of separation quality, aiming to retain the relative ranking of silhouettes while improving scalability. The method also allows tuning between macro- and micro-averaged silhouette objectives and explores two weighting formulations. The authors validate K-Sil on several synthetic and real-world datasets, showing consistent improvements over standard k-means, though most experiments appear to involve moderate-sized or low-dimensional data rather than very high-dimensional or large-k scenarios.

**Strengths:**

This paper proposes an interesting extension of k-means that uses silhouette information to weight data points. The idea is simple and well-motivated, aiming to improve separation and reduce the impact of noise. The method is clearly explained and easy to follow, with a good balance between intuition and technical detail. The authors provide both theoretical analysis and experimental results to support their claims. The macro- and micro-silhouette objectives, together with the approximate silhouette computation, make the approach flexible and more efficient. Overall, it is a clear and thoughtful contribution that could be useful in many clustering settings.

**Weaknesses:**

The paper layout seems slightly off — the text column is narrower than the standard ICLR template (see lines 130 and 137), leaving extra unused space.

Figure 1 compares K-Sil and standard k-means using a single initialization; this should be repeated over multiple runs and reported using quantitative metrics such as average silhouette score and quantization error for a fair comparison.

The algorithm introduces additional computational overhead relative to k-means, and its dependence on approximate silhouette estimates and new parameters (exponential/power weights) raises questions about numerical stability and reproducibility.

The experimental section would be stronger with larger or higher-dimensional datasets, as the current benchmarks (e.g., 20 Newsgroups) are relatively small.

Finally, performance is evaluated mainly on the macro-silhouette score—the same metric the method optimizes. Including comparisons using standard k-means objectives, such as total within-cluster variance or variance-ratio criteria, would provide a fairer assessment of the practical benefits.

**Questions:**

What is the runtime increase of the algorithm relative to standard k-means?
Can you add run time on Tables 1 and 2?

---

### Note · Authors · 2025-12-01

**Comment:**

We have decided to withdraw this submission. We thank the Area Chair and reviewers for their time and effort.

**Withdrawal Confirmation:**

I have read and agree with the venue's withdrawal policy on behalf of myself and my co-authors.